# A Cheaper and Better Diffusion Language Model with Soft-Masked Noise

**Jiaao Chen**[†*]**, Aston Zhang**[†]**, Mu Li, Alex Smola, Diyi Yang**[◇]
[†]Georgia Institute of Technology, [‡]Meta GenAI, [◇]Stanford University

## Abstract

Diffusion models that are based on iterative denoising have been recently proposed and leveraged in various generation tasks like image generation. Whereas, as a way inherently built for continuous data, existing diffusion models still have some limitations in modeling discrete data, e.g., languages. For example, the generally used Gaussian noise can not handle the discrete corruption well, and the objectives in continuous spaces fail to be stable for textual data in the diffusion process especially when the dimension is high. To alleviate these issues, we introduce a novel diffusion model for language modeling, Masked-Diffusion LM, with lower training cost and better performances, inspired by linguistic features in languages. Specifically, we design a linguistic-informed forward process which adds corruptions to the text through strategically soft-masking to better noise the textual data. Also, we directly predict the categorical distribution with cross-entropy loss function in every diffusion step to connect the continuous space and discrete space in a more efficient and straightforward way. Through experiments on 5 controlled generation tasks, we demonstrate that our Masked-Diffusion LM can achieve better generation quality than the state-of-the-art diffusion models with better efficiency. Code is available at `https://github.com/SALT-NLP/Masked_Diffusioin_LM`.

## 1 Introduction

We present a novel diffusion method for modeling languages, Masked-Diffusion LM (language model), which uses strategic soft-masking informed by linguistic features to corrupt both the discrete and continuous space, and then iteratively denoise them back by predicting the categorical distribution. Specifically, a strategic soft-masking process is designed that gradually adds perturbation to the input text in an order from harder or

more informative words to simpler or less informative words through soft-masking. As a result, the models are encouraged to recover and generate the text following an *easy-first-generation* nature (Dieleman et al., 2022) to improve the generation structure and quality with more flexibility. Also, during the diffusion process, we directly predict the discrete token with cross-entropy loss that maps the continuous space to discrete textual space to stabilize the intermediate diffusion steps. Through our proposed Masked-Diffusion LM, the application-specific performance metrics as well as training efficiency are significantly improved over current diffusion language models based on experiments.

Our work is inspired by recent advances in diffusion models (Sohl-Dickstein et al., 2015; Ho et al., 2020; Song et al., 2021; Yang et al., 2022; Ramesh et al., 2022; Rombach et al., 2022) that are introduced as a new generative modeling approach based on iterative denoising and have achieved high-quality generations for visual and audio modalities (Ramesh et al., 2022; Rombach et al., 2022; Saharia et al., 2022; Nichol and Dhariwal, 2021; Kong et al., 2020).

Although these approaches have received growing attention and achieved impressive success, applying diffusion models to textual domain is still challenging and under-explored due to the discrete nature of the text (e.g., one-hot vectors) compared to continuous data like images (e.g., RGB values) (Li et al., 2022). A few prior works (Li et al., 2022; Gong et al., 2022; He et al., 2022; Austin et al., 2021; Hoogeboom et al., 2021b) that explore using diffusion models on textual data can be divided into two lines. The first is to extend diffusion models to discrete state spaces (Austin et al., 2021; Hoogeboom et al., 2021b,a). The second is to perform the diffusion process and its reverse process in the continuous domain and bridge the continuous and the discrete domain through embedding and rounding (Li et al., 2022; He et al., 2022), for example,

---

*Correspondence to Jiaao Chen <jiaaochen@gatech.edu> and Aston Zhang <az@astonzhang.com>.

Diffusion-LM (Li et al., 2022). Despite the improvements, most previous works fail to leverage the linguistic features (e.g., words in sentences are with different importance) to noise the input textual data and recover it back in a more suitable way. Besides, they usually neglect or fail to adapt large pre-trained language models (PLMs) (Devlin et al., 2019; Liu et al., 2019; Yang et al., 2019; Joshi et al., 2019; Sun et al., 2019; Clark et al., 2019; Lewis et al., 2020; Bao et al., 2020; He et al., 2020; Raffel et al., 2020), which is an unmissable treasure in the NLP community: their adopted $k$-nearest-neighbor rounding technique that maps continuous space to discrete space cannot handle high-dimensional data in a stable and efficient way (Li et al., 2022). As a result, a corruption process tailored for languages and the objective that allows efficient and straightforward discrete and continuous space transformation is in great need. Our Masked-Diffusion LM realizes this extension.

To demonstrate the effectiveness of our introduced Masked-Diffusion LM, we perform experiments on E2E dataset (Novikova et al., 2017) and 5 controllable generation tasks (Li et al., 2022) including Semantic Content, Parts-of-speech, Syntax Tree, Syntax Spans, and Length. We observe that our Masked-Diffusion LM can (i) achieve the state-of-the-art performances compared to recent baseline models, and (ii) allow more efficient training and inference compared to previous Diffusion-LM.

To summarize, our contributions are: (1)We introduce a strategic masking noise strategy guided by linguistic features to corrupt the textual data in diffusion models for modeling languages. (2) We use linear layers and cross-entropy objectives to bridge the continuous and discrete spaces in the diffusion process for efficiency and stability. (3) We conduct experiments on different controllable generation tasks to demonstrate the effectiveness of our proposed methods compared to previous diffusion language models.

## 2  Related Work

**Diffusion Models for Language**   There has been growing attention in deep generative diffusion models, which is a latent variable generative method based on iterative denoising (Sohl-Dickstein et al., 2015; Ho et al., 2020; Song et al., 2021). Through a forward and diffusion process, diffusion models have shown state-of-the-art sample quality on generating in the continuous domain such as producing

images and audio (Ramesh et al., 2022; Rombach et al., 2022; Kong et al., 2020; Savinov et al., 2022). Despite their huge success, it is still challenging and under-explored to adapt diffusion models to discrete domains like languages. A few recent works have modified the diffusion models for textual data. For example, *discrete* forward processes, such as categorical transition kernels (Hoogeboom et al., 2021a; Ye et al., 2023), uniform transition kernels, and absorbing kernels (Hoogeboom et al., 2021b), have been introduced. However, replacing continuous diffusion with a discrete corruption process affords some flexibility (Dieleman et al., 2022; Zheng et al., 2023; Reid et al., 2022). Other works have also made efforts to model text in the *continuous* embedding space and applied Gaussian noise uniformly to every token (Li et al., 2022; He et al., 2022; Chen and Yang, 2023), which is closer to the settings in previous works of diffusion models. However, they neglect the inherent linguistic features in the text (e.g., *different words are playing different roles in sentences*) so the generated text often lacks coherence (He et al., 2022). Besides, the $k$-nearest-neighbor rounding technique (Li et al., 2022; Gao et al., 2022) holds up the decoding and convergence speed especially when the vocabulary is large or the hidden dimension is high, thus limiting the potential of combining large pre-trained language models (Devlin et al., 2019; Liu et al., 2019; Yang et al., 2019; Joshi et al., 2019; Sun et al., 2019; Clark et al., 2019; Lewis et al., 2020; Bao et al., 2020; He et al., 2020; Raffel et al., 2020). To alleviate these issues, in our work, we introduce a linguistic-informed soft-masking process to corrupt the discrete and continuous space with structures, and then use linear projections and cross-entropy objectives to directly map the latent variables to textual data for better efficiency and generating better text.

**Non-Autoregressive Text Generation**   Most language models (Chowdhery et al., 2022; Brown et al., 2020) and text generation models (Vaswani et al., 2017a; Eikema and Aziz, 2021; Chen and Yang, 2020, 2021) follow a left-to-right autoregressive manner. However, the fixed generation order prevents the models' flexibility in editing former text based on later generation results, especially for global controllable generation settings. To overcome the limitations, non-autoregressive text modeling has been proposed (Ghazvininejad et al., 2019; Ren et al., 2020; Gu et al., 2018; Sa-

haria et al., 2020; Savinov et al., 2022) through masked language models (Ghazvininejad et al., 2019), iterative sequence alignment (Saharia et al., 2020), insertion and deletion (Gu et al., 2018), or unrolling the generation path (Savinov et al., 2022). Our Masked-Diffusion LM achieves the non-autoregressive generation through gradually recovering the intermediate latent variables in a planned sequence from the forward process.

**Plug-and-Play Controllable Generation**  Our work is also closely related to the line of research about plug-and-play controllable generation methods (Yang and Klein, 2021; Dathathri et al., 2020; Krause et al., 2021; Liu et al., 2021), which modify the outputs based on extra guidance such as classifiers without changing or fine-tuning the pre-trained language models. Dathathri et al. (2020) used gradients to edit the autoregressive language model's hidden representations to fulfill the control guidance. Yang and Klein (2021) proposed to reweight the predicted token from the language models while (Krause et al., 2021; Liu et al., 2021) further fine-tuned a smaller LM to reweight the token predictions. In this work, we apply the gradient-based plug-and-play approach to our Masked-Diffusion LM for controllable generation by making classifier-guided gradient updates to the intermediate latent variables during the diffusion.

## 3  Method: the Masked-Diffusion LM

In this section, we describe our introduced Masked-Diffusion LM. The overall diagram is shown in Figure 1 and Algorithm 1,2. Different from the recent diffusion models for languages, e.g., Diffusion-LM (Li et al., 2022), which are based on continuous diffusion models, we propose to make corruptions in both discrete and continuous space to help modeling the textual data. Specifically, we formulate a novel corruption process as an alternative to Gaussian diffusion (in Section 3.2) and we directly map continuous vectors to discrete inputs in every diffusion step with cross-entropy objectives (in Section 3.3). Moreover, our approach could easily integrate pre-trained language models (in Section 3.4).

### 3.1  Embedding

For the input sentence $d$ with $l$ tokens $d = \hat{w}_{1:l}$, we first map the discrete tokens to the continuous space and form the initial latent variable, $X_0$, through a learnable embedding layer or an encoder $e(.)$:

$$X_0 = w_{1:l} = e(w_{1:l}). \tag{1}$$

This bridges the discrete space and continuous space. We will then add designed soft-masked noise to the tokens' representations in the later diffusion models.

### 3.2  Forward Process with Soft-Masking

Different words in sentences play different roles. As a result, when corrupting the sentences and recovering the sentences, words with various importance should be treated differently. Thus, in this work, instead of evenly adding Gaussian noise to all the token embeddings like in Diffusion-LM (Li et al., 2022), we add soft-masked noise to different tokens in the input text in different stages to corrupt the text gradually with structures. Intuitively, more important words would be perturbed with soft-masks in an earlier stage so that the model could be encouraged to generate them in the later phase to follow the *easy-first-generation* nature of language planning and generation.

In this work, we consider the following aspects to measure and define the importance of words in one sentence:

**Word Relevancy**  We use the tf-idf weights (Dessí et al., 2020), $w_{\text{tf-idf}}$, of the word as one way to measure the relevance of word $w$ in one sentence $d$:

$$
\begin{aligned}
w_{\text{tf-idf}}(w, d) = \frac{f_{w,d}}{\sum_{w' \in d} f_{w',d}} \\
\log \frac{N}{1 + |\{d \in D : w \in d\}|},
\end{aligned}
\tag{2}
$$

where the $f_{w,d}$ is the number of times that word $w$ occurs in sentence $d$, $N$ is the number of sentences in the corpus, and $D$ is the set of sentences, and $|\{d \in D : w \in d\}|$ is number of sentences where the word $t$ appears. A higher weight for word $w$ in sentence $d$ in tf–idf means that the word might be more important in the sentence.

**Entropy**  We also consider measuring the amount of information with entropy $H$ (Bentz and Alikaniotis, 2016; He et al., 2022) in the word $w$ to reflect the importance of that word:

$$H(w) = -p(w) \log(p(w)) \tag{3}$$

where $p(w) = \frac{f_w}{\sum_{j=1}^{V} f_j}$ represents the probability of word $w$ and $f$ is the word Reluency in the corpus. A word with lower entropy indicates that the word might contain less information and thus be

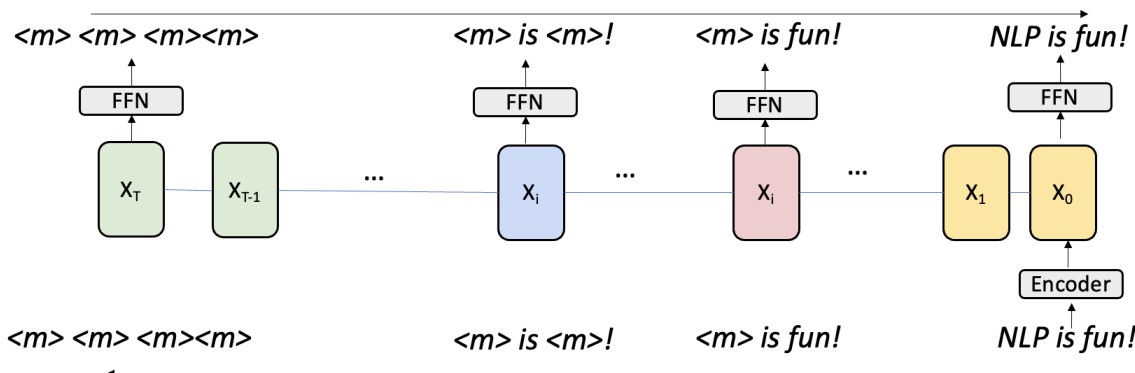

**Diffusion Process** $p_\theta(\mathbf{x}_{t-1} \mid \mathbf{x}_t, t)$

**Forward Process** $q(\mathbf{x}_t \mid \mathbf{x}_{t-1})$

Figure 1: The overall process of our Masked-Diffusion LM. In the forward process, soft-mask is added to more informative words earlier to gradually corrupt the input text. For example, *NLP* is soft-masked prior to stop words like *is*. Then in the diffusion process, models learn to generate easy words like *is* first and then fill in more important words such as *fun* and *NLP*.

less important compared to the words with higher entropy.

In practice, we combine these two measures (with normalization) to decide the importance $I$ of the word $w$ in one sentence $d$ by:

$$I(w) = \frac{x_{\text{tf-idf}}(w, d)}{\sum_{w' \in d} w_{\text{tf-idf}}(w', d)} + \frac{H(w)}{\sum_{w' \in d} H(w')}. \quad (4)$$

Based on the introduced importance $I$ of the words in a sentence, we first divide these words into $m$ buckets $\{W_{1:m}\}$. The buckets with lower indices include words with higher importance. We will add soft-masked noise to words with higher importance before words with lower importance. By doing this, models could learn to generate the easier words first and then generate harder words in the reversed denoising process for better generation quality. Specifically, at every step $t$, we will add a small amount of Gaussian noise to the hidden representation of the word $w_i$ in bucket $W_{\lfloor \frac{tm}{T} \rfloor}$:

$$q(w_{i,t+1}|w_{i,t}) = N(w_{i,t+1}; \sqrt{(1 - \beta_t)}w_{i,t}, \beta_t I), \quad (5)$$

where $\beta_t$ is the amount of noise added at diffusion step $t$.

We further apply a square-root noise schedule following Li et al. (2022) to gradually increase $\beta_t$:

$$\beta_t = 1 - \sqrt{t/T + s}, \quad (6)$$

---

**Algorithm 1** Forward Process

**Input** A sentence $X = [x_0, \ldots, x_n]$.
**Output** Corrupted hidden representations $H_T = [h_0, \ldots, h_n]$.
1: Encode the sentence into hidden representations via an encoder $e(.)$: $H_0 = e(X)$.
2: **for** $t = 1, \ldots, K$ **do**
3:     Add soft-masking noise to $H$ based on the importance of tokens (from higher-importance to lower-importance): $H_{t+1} = $ soft-masking$(H_t)$
4: **end for**

---

where $s$ is a small constant that corresponds to the starting noise level. Thus, less noise would be added to harder words to stabilize the training. By performing the above noising steps, initial latent variable $X_0$ is gradually corrupted to a series of noisy latent variables $X_{1:T}$.

### 3.3 Diffusion Process

After the forward process to corrupt the input tokens in sentences $d$ into latent variables $X_{1:T}$, we then gradually denoise $X_T$ back to $X_0$ through diffusion steps, $\hat{X}_{t-1} = p(\hat{X}_t|\theta)$, where $\theta$ is the learned parameter to model the state transition. In practice, we model the transition with Transformers (Vaswani et al., 2017b).

After every diffusion step $t \in (0, T]$, instead of minimizing the distance between the hidden rep-

---

**Algorithm 2** Diffusion Process

**Input** Corrupted hidden representations $H = [h_0, \ldots, h_n]$.
**Output** A sentence $X = [x_0, \ldots, x_n]$.

1: Utilize a transition network $f(.)$ to recover the last state: $H_{t-1} = f(H_t)$
2: Utilize a linear layers to map hidden representations to actual tokens $X_{t-1} = g(H_{t-1})$
3: Compute the loss $\mathcal{L}_t$ and update the transition network.
4: Do the above steps until it recovers the sentence.

---

resentations of $\hat{X}_{t-1}$ and $X_0$ (Li et al., 2022), we first directly map the continuous space to discrete space using a learnable linear layer $f(.)$ and then minimize a weighted cross entropy between the predicted sentence and (i) the original sentence $d$ and (ii) the masked sentence $\hat{d}$ at time step $t-1$:

$$\mathcal{L}_t = \gamma_t CE(f(\hat{X}_{t-1}), d; \theta) \\ + (1 - \gamma_t) CE(f(\hat{X}_{t-1}), \hat{d}; \theta), t \in (0, T]$$

Here, $\gamma_t = \frac{T-t}{T}$. In other words, we put higher weights on the masked tokens that are masked in this time step during the forward process and put lower weights to the other tokens. So the models are learned to generate the corresponding masked tokens first at every time step.

### 3.4 Adapting Pre-trained Language Models

Our introduced Masked-Diffusion LM also allows the use of large pre-trained language model (Devlin et al., 2019; Liu et al., 2019; Yang et al., 2019; Joshi et al., 2019; Sun et al., 2019; Clark et al., 2019; Lewis et al., 2020; Bao et al., 2020; He et al., 2020; Raffel et al., 2020). In this work, we use BERT (Devlin et al., 2019) as an example. To combine the prior knowledge in large language models, it is straightforward to directly replace the embedding layer $e(.)$ with the pre-trained model and use the pre-trained model to get the hidden representations of input tokens as the initial state in diffusion models. We use the final linear layers in pre-trained models to predict the tokens. For efficiency, in our experiments, when using pre-trained models, we freeze the parameters in them and only learn the transition model $\theta$ in our Masked-Diffusion LM.

## 4 Controllable Text Generation with Masked-Diffusion LM

In this section, we illustrate how we apply our Masked-Diffusion LM to fulfill controllable text generation. Inspired by recent plug-and-play methods (Yang and Klein, 2021; Dathathri et al., 2020; Krause et al., 2021; Liu et al., 2021), we conduct controls $c$ from external modules (e.g., classifiers) directly on the latent variables $X_t$ in every intermediate step $t \in [0, T]$ in our Masked-Diffusion LM:

$$p(X_{0:T} \mid c) = \prod_{t=1}^{T} p(X_{t-1} \mid X_t, c). \quad (7)$$

We follow the conditional independence assumption (Yang and Klein, 2021; Dathathri et al., 2020; Krause et al., 2021; Liu et al., 2021) and decompose the above joint probability into a sequence of control task at every time step $t$:

$$p(X_{t-1} \mid X_t, c) \propto p(X_{t-1} \mid X_t) \cdot p(c \mid X_{t-1}, X_t) \\ = p(X_{t-1} \mid X_t) \cdot p(c \mid X_{t-1}). \quad (8)$$

As a result, for the $t$-th step, we run gradient updates on $X_t$ to generate $X_{t-1}$:

$$\nabla_{X_{t-1}} \log p(X_{t-1} \mid X_t, c) = \lambda \nabla_{X_{t-1}} \\ \log p(X_{t-1} \mid X_t) + \nabla_{X_{t-1}} \log p(c \mid X_{t-1}), \quad (9)$$

where both $\log p(X_{t-1}|X_t)$ and $\log p(c|X_{t-1})$ are differentiable: the first term is parametrized by the transition Transformers, $\theta$, in Masked-Diffusion LM, and the second term is parametrized by extra neural network classifiers. Note that the extra classifiers are trained with the diffusion latent variables as input to allow direct gradient updates on the latent space. Note that $\lambda$ is a fluency regularization hyper-parameter to balance the fluency (gradient updates from Masked-Diffusion LM) and control (gradient updates from classifiers) in order to further improve the generation quality.

For the decoding strategy, following Li et al. (2022), the Minimum Bayes Risk (MBR) decoding (Kumar and Byrne, 2004) is used to aggregate and select the sample that has the lowest expected loss under the specified loss function from the Masked-Diffusion LM.

## 5 Experiments

### 5.1 Datasets

In this work, we train our Masked-Diffusion LM on the E2E datasets (Novikova et al., 2017), which

| Methods | Semantic Content | | POS | | Syntax Tree | | Syntax Spans | | Length | |
|---|---|---|---|---|---|---|---|---|---|---|
| | Acc | Fluency | Acc | Fluency | Acc | Fluency | Acc | Fluency | Acc | Fluency |
| PPLM | 9.9 | 5.32 | - | - | - | - | - | - | - | - |
| FUDUGE | 69.9 | 2.83 | 27.0 | 7.96 | 17.9 | 3.39 | 54.2 | 4.03 | 46.9 | 3.11 |
| Diffusion-LM | 81.2 | 2.55 | 90.0 | 5.16 | 86.0 | 3.71 | 93.8 | 2.53 | 99.9 | 2.16 |
| + BERT | 77.4 | 2.68 | 86.2 | 5.43 | 82.3 | 3.92 | 89.3 | 3.13 | 99.9 | 2.68 |
| Masked-Diffusion LM † | 81.9 | 2.35 | 91.6 | 5.03 | 86.6 | 3.66 | 94.7 | 2.48 | 99.9 | 2.13 |
| + BERT † | **82.9** | **2.30** | **92.9** | **4.78** | **89.7** | **3.44** | **95.8** | **2.33** | **100** | **2.08** |

Table 1: Main Results. The Accuracy (↑) and the Fluency (↓) of different methods on five controllable generation tasks including semantic content, POS, syntax tree, syntax spans and length. † indicates our methods.

| Methods | Training (h) | Inference (s) |
|---|---|---|
| Diffusion-lm | 8.0 | 80 |
| +BERT | 15.2 | 920 |
| Masked-Diffusion LM | 3.4 | 68 |
| +BERT | 4.8 | 700 |

Table 2: Training time and inference time (generating 50 samples) for different models.

consists of 50K restaurant reviews together with the labels in terms of food type, price, and customer ratings.

Following Li et al. (2022), we conduct 5 control tasks to evaluate the learned Masked-Diffusion language model:

- **Semantic Content.** For a given field (e.g., *food*) and value (e.g., *Japanese*), sentences that covers field=value need to be generated. We evaluate the accuracy of the generated sentence by examine the exact match rate of "value" (word mention).

- **Parts-of-speech.** For a given sequence of parts-of-speech (POS) tags (e.g., *Noun Verb Determiner Noun*), the models need to produce the sentence with the same length and follow the exact given POS tag sequence (e.g., *Birds eat the warms*). We evaluate the accuracy of the generation by checking the word-level POS tag exact match (under an oracle POS tagger).

- **Syntax Tree.** For a given syntactic parse tree, the generated sentence should have the same parse tree. We evaluate the accuracy by first parsing the generated sentence with an off-the-shelf parser and report the F1 scores compared to the given parse.

- **Syntax Spans.** For a given (span, syntactic category) pair (e.g., *(2, 5, VP)*), the parse tree

of the generated sentence should match the given syntactic category over the given spans. We evaluate the accuracy of the sentence by the exact match rate of the given spans.

- **Length.** For a given target length (e.g., *20*), the models need to generate a sentence within ±2 of the given target. We evaluate the accuracy by the match rate of the sentence lengths.

For every control task, we sample 200 control targets $c$ from the validation splits, and we generate 50 samples for each control target. The first four tasks rely on a classifier to guide the diffusion, and the last one task is classifier free. To further evaluate the fluency of the generated sentences from models, we use a teacher LM (i.e., a carefully fine-tuned GPT-2 model) and report the perplexity of generated text under the teacher LM. A lower perplexity indicates better sample quality and fluency.

### 5.2 Baselines

We compare our Masked-Diffusion LM with the following state-of-the-art baselines on controllable generation tasks:

- **PPLM** (Dathathri et al., 2020) runs gradient ascent on the pre-trained language models' hidden representations to increase the classifier probabilities and language model probabilities.

- **FUDGE** (Yang and Klein, 2021) reweights the predicted tokens from the pre-trained language models by a discriminator which takes in a prefix sequence and predicts whether the complete sequence would satisfy the constraint.

- **Diffusion-LM** (Li et al., 2022) learns an embedding to map discrete text into the continuous space where it performs Gaussian

| Methods | Semantic Content | POS | Syntax Tree | Syntax Spans | Length |
|---|---|---|---|---|---|
| Diffusion-lm | 2.89 | 2.76 | 3.16 | 2.88 | 2.46 |
| +BERT | 3.87 | 3.46 | 3.72 | 3.68 | 3.34 |
| Masked-Diffusion LM | 2.56 | 2.48 | 2.88 | 2.35 | 2.18 |
| +BERT | **1.32** | **1.28** | **1.16** | **1.55** | **1.86** |

Table 3: The average ranking every method receives from human evaluation (lower is better).

| Noise Type | Semantic Content | |
|---|---|---|
| | Acc | Fluency |
| Gaussian | 75.3 | 3.01 |
| Random Mask | 78.8 | 2.67 |
| Mask w. POS | 80.4 | 2.58 |
| Mask w. Entropy | 81.1 | 2.44 |
| Mask w. Rel | 80.8 | 2.52 |
| Mask w. Entropy+Rel † | **81.6** | **2.38** |

Table 4: Performances on Semantic Content of Masked-Diffusion LM with different types of noise applied in forward noising process. † indicates our method.

diffusion process. Also, a rounding step is designed to map the embeddings back into discrete texts. For every control task, the Diffusion-LM infuses the controlling signals in every diffusion step.

## 5.3 Experimental Setting

We use a Transformer with 80M parameters to parameterize our Masked-Diffusion LM, with a sequence length $n = 64$, diffusion steps $T = 500$, and a square-root noise schedule. For Masked-Diffusion LM, we set the hidden dimension to 128. We set the number of word buckets $m = 3$. When combining pre-trained models, we incorporate BERT-base (Devlin et al., 2019) with about 110M parameters. We use BERT to encode the input text into vectors with dimension of 768 and freeze the parameters in BERT. We learn Masked-Diffusion LM with the AdamW optimizer (Loshchilov and Hutter, 2019) for 20,000 steps with learning rate of 3e-4, dropout probability of 0.1, and batch size of 32. We use a linear warmup schedule starting with 1,000 warmup steps. All experiments are conducted on NVIDIA A100 Tensor Core GPUs. We use 4 GPUs for training and a single GPU for sampling.

## 5.4 Results

We show the main results on five controllable generation tasks in Table 1. When the diffusion process is engaged, the performances on all the controlled generation tasks receives significant boosts (e.g., 81.2 of Diffusion-LM vs. 69.9 if FUDUGE on Semantic Content task), suggesting the superiority of the diffusion model on controllable generation tasks. While the previous Diffusion-LM can not be well combined with large language model like BERT (e.g., a 5% drop on Semantic Content accuracy), largely due to the fact that their way (rounding) to bridge continuous space and discrete space suffers from significantly higher dimensions. Compared to Diffusion-LM, our proposed Masked-Diffusion LM consistently outperforms the previous models in all tasks (e.g., a 1.7% improvement on the POS task), indicating the effectiveness of our introduced linguistic-informed noise forward process. Also, when combined with large language models like BERT, our method significantly outperforms the previous methods, demonstrating that our approach can be well aligned with pre-trained models.

**Efficiency** We also display the training cost and inference cost in Table 2. Compared to the previous Diffusion-LM, our method requires significantly less training time to converge and needs less inference time to generate sentences. This is because our introduced noise process is more stable and suitable for modeling languages. Besides, the objectives we introduced are more efficient than the rounding techniques in previous work.

**Human Evaluation** We then conduct human evaluation to evaluate the generated conversations qualitatively. We ask native speakers of English from Amazon Mechanical Turk to rank the quality of 50 generated sentences (randomly sampled) from different models for every control task. Specifically, annotators need to rank different system outputs based on the (i) fluency (whether the

| Methods | Semantic Content | | POS | | Syntax Tree | | Syntax Spans | | Length | |
|---|---|---|---|---|---|---|---|---|---|---|
| | Acc | fluency | Acc | fluency | Acc | fluency | Acc | fluency | Acc | fluency |
| L2 | 81.1 | 2.44 | 90.6 | 5.17 | 86.2 | 3.68 | 94 | 2.51 | 99.8 | 2.14 |
| L2-BERT | 80.1 | 2.48 | 89.4 | 5.82 | 84.1 | 3.91 | 93.2 | 2.88 | 99.9 | 2.89 |
| CE † | 81.9 | 2.35 | 91.6 | 5.03 | 86.6 | 3.66 | 94.7 | 2.48 | 99.9 | 2.13 |
| CE-BERT † | **82.9** | **2.30** | **92.9** | **4.78** | **89.7** | **3.44** | **95.8** | **2.33** | **100** | **2.08** |

Table 5: Performances of Masked-Diffusion LM trained with different objectvies on controllable generation tasks. † indicates our method.

| Case Study | Sentences |
|---|---|
| Input | *7* |
| $t = 500$ | [*mask*] [*mask*] [*mask*] [*mask*] [*mask*] [*mask*] [*mask*] |
| $t = 400$ | [*mask*] is an [*mask*] restaurant . |
| $t = 200$ | The [*mask*] is an Indian restaurant . |
| $t = 0$ | The Mill is an Indian restaurant . |
| Input | *name : Travellers Rest Beefeater* |
| $t = 500$ | [*mask*] [*mask*] [*mask*] [*mask*] [*mask*] [*mask*] [*mask*] [*mask*] [*mask*] [*mask*] [*mask*] [*mask*] |
| $t = 400$ | [*mask*] Rest [*mask*] is a [*mask*] [*mask*] [*mask*] that is [*mask*] . |
| $t = 200$ | Travellers Rest [*mask*] is a reasonably [*mask*] restaurant that is awesome . |
| $t = 0$ | Travellers Rest Beefeater is a reasonably priced restaurant that is awesome . |

Table 6: Examples of the intermediate generated text of our Masked-Diffusion LM on the Length and Semantic Content tasks.

given sentence is readable and fluent) and (ii) the controllability (whether the given sentence match the given control conditions). To increase annotation quality, we require turkers to have a 98% approval rate with over 10,000 approved tasks for their previous work. The pay rate was $0.15 per hit. Every example is assessed by 3 annotators, and the rank for every sentence is aggregated by majority voting. The Intra-Class Correlation (*ICC1k*) was 0.63, indicating moderate agreement (Koo and Li, 2016). The results are shown in Table 3. As it shows, our proposed Masked-Diffusion LM and its variation with BERT received the best average ranks, suggesting the effectiveness of our proposed diffusion modeling strategy for languages.

## 5.5 Ablation Studies

We then perform ablation studies to demonstrate the effectiveness of our introduced linguistic-informed noise and the cross entropy objectives.

**Noise Strategy** We first demonstrate the performances on Semantic Content task of Masked-Diffusion LM with different types of noise strategy in Table 4. *Gaussian* adds Gaussian noise to all the tokens in the input sentence in the forward process following Li et al. (2022). We also compare different masking noise strategies: (i) Random Mask, where the soft-mask is added to tokens in a random

order. (ii) Mask with POS, where the soft-mask perturbs the tokens in an order (noun → verb → other words) based on POS tags. Our introduced noise strategy (Mask with Entropy and Reluency) shows significantly better performances on semantic content generation. This indicates that our introduced noise strategy that considers the linguistic features in sentences is providing more appropriate perturbation to the textual data for the diffusion process.

**Objectives** We further show the impact of different objectives in Table 5. We compare our used cross entropy objectives with the $L_2$ object that is used in Li et al. (2022) where they minimize the distance between latent intermediate variables and the initial latent variable instead of directly predicting the text. We observe that cross entropy objectives slightly perform better than $L_2$ when the pre-trained model is not used. After combining with large language models, CE-BERT significantly outperforms the $L_2$-BERT, indicating the effectiveness of our introduced objectives in terms of incorporating large language models.

## 5.6 Case Studies

We also include some examples of intermediate steps of Masked-Diffusion LM in Table 6. In the denoising diffusion process, easy words are generated first. For example, "*is*", "*an*", and "*restaurant*".

With more diffusion steps, sentences are enriched with more informative words such as "*Mill*" and "*Indian*". It shows that our Masked-Diffusion LM encourages the generation to follow an easy-first order for stable and better generation quality.

## 6 Conclusion

In this work, we present a novel diffusion model for language, Masked-Diffusion LM, which corrupts the discrete text with a linguistic-informed soft-masking strategy and then iteratively denoises them back by directly predicting the text. Specifically, we gradually soft-mask the tokens in the sentence following an order from more informative words to less informative words in the forward process. This satisfies the flexibility for diffusion models, as well as encourages the easy-first-generation nature in the denoising process for better generation quality. Also, we directly predict the discrete token during the diffusion process with the cross-entropy loss to stabilize the intermediate diffusion steps and make our approach orthogonal to large pre-trained language models. Experiments on E2E dataset and five controllable generation tasks including Semantic Content, Parts-of-speech, Syntax Tree, Syntax Spans, and Length show that our Masked-Diffusion LM can (i) achieve the state-of-the-art performances compared to recent baseline models and (ii) allow more efficient training and inference compared to the previous Diffusion-LM.

## 7 Limitations

In this work, we mainly leverage linguistic soft-masking such as word relevancy and word entropy. We encourage future work to explore how to incorporate other linguistic structures to design the nosing process. And we mainly test with smaller models like simple transformer models as well as BERT-based models. Future work might test with larger pre-trained models to evaluate whether diffusion methods would work better or not. Also, we focused on controllable generation to evaluate the models. Future work may study different downstream tasks.

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
