# OpenReview forum: "A Cheaper and Better Diffusion Language Model with Soft-Masked Noise"
_EMNLP/2023/Conference — EMNLP 2023 Main_

### Official Review · Reviewer_4Jws · 2023-08-04

**Soundness:** 4

**Excitement:**

4: Strong: This paper deepens the understanding of some phenomenon or lowers the barriers to an existing research direction.

**Paper Topic And Main Contributions:**

This paper proposed a novel way of adding noise during the forward process of the diffusion language model, which first proposes a method to compute the importance of each word in a sentence, then weigh the noise that is added into word embedding by the corresponding importance of the word. Meanwhile, this paper proposes a simple and effective way to alleviate the discrete corruption problem.

**Questions For The Authors:**

1. Do the authors try to build this method upon LLM?

**Reasons To Accept:**

1. The proposed method for computing the importance of the word in a sentence is very interesting, and combining the importance of the word with the forward process is very ingenious.

2. Mapping the continuous space to discrete space is very simple and elegant for alleviating the problem led by discrete corruption during the diffusion process.

3. The analysis and comparison, like in Table 4, sufficiently show the effectiveness of the proposed approach.

**Reasons To Reject:**

I do not see any risk.

**Reproducibility:**

4: Could mostly reproduce the results, but there may be some variation because of sample variance or minor variations in their interpretation of the protocol or method.

**Reviewer Confidence:**

4: Quite sure. I tried to check the important points carefully. It's unlikely, though conceivable, that I missed something that should affect my ratings.

---

> ### Author Rebuttal · Authors · 2023-08-28
>
> Thanks for your positive assessments.
>
> ### On the questions about the use of LLM
> Yes, our methods can be easily built upon LLMs like BERT as stated in the Section 4.4 and Table 4. We could just replace the encoder (shown in Figure 1) with any LLMs to map the discrete tokens into continuous hidden representations. Then we applied the soft-masked corruption process and the reverse process to reconstruct the sequence. In the reverse process, we learn the transition network from a randomly initialized transformer based on the token predictions.

---

### Official Review · Reviewer_YAfc · 2023-08-04

**Soundness:** 3

**Excitement:**

2: Mediocre: This paper makes marginal contributions (vs non-contemporaneous work), so I would rather not see it in the conference.

**Paper Topic And Main Contributions:**

This paper presents a novel diffusion model for generating text, called Masked-Diffusion LM. This model has several key contributions. Firstly, it uses a linguistic-informed soft masking process during the forward diffusion process. This means that important words, based on TF-IDF and entropy, are masked earlier, so that the model learns to generate them later. This encourages an easy-first generation. Secondly, it employs a cross-entropy loss to directly predict tokens and map the continuous latent space to discrete tokens during diffusion. This is more efficient than previous rounding techniques. Finally, Masked-Diffusion LM achieves state-of-the-art performance compared to prior diffusion LMs in experiments on controllable text generation tasks, such as enforcing semantic content and POS tags. It also allows for more efficient training and inference.

In summary, this paper proposes a linguistically-motivated masking scheme for the text corruption process in diffusion models and a direct discrete token prediction technique during diffusion that improves efficiency. Evaluation on controllable generation tasks demonstrates improved performance over prior diffusion LMs for text.

**Reasons To Accept:**

Here are some of the main strengths and benefits of this paper:

- Novel diffusion modeling technique for text: The proposed masking and discrete prediction scheme provides a new way to adapt diffusion models to better suit textual data. This contributes to advancing diffusion models for NLP.

- Improved controllable text generation: The experiments demonstrate state-of-the-art performance on several controllable text generation tasks. This is a useful generation capability for applications.

- The proposed techniques allow faster training and inference compared to prior diffusion LMs, improving efficiency.

- Well-written: The paper is clearly written and provides sufficient background and explanation of the new techniques. This makes it accessible to the NLP community.

**Reasons To Reject:**

Here are some points to keep in mind:

- No comparison to other discrete diffusion techniques: The paper compares mainly to continuous diffusion methods for text. Including comparisons to other discrete techniques could offer more informative insights.

- Pre-trained model dependencies: The benefits of this paper rely on transformer-based pre-trained models. It may be helpful to test the robustness of different model choices.

**Reproducibility:**

2: Would be hard pressed to reproduce the results. The contribution depends on data that are simply not available outside the author's institution or consortium; not enough details are provided.

**Reviewer Confidence:**

3: Pretty sure, but there's a chance I missed something. Although I have a good feel for this area in general, I did not carefully check the paper's details, e.g., the math, experimental design, or novelty.

---

> ### Author Rebuttal · Authors · 2023-08-28
>
> Thanks for your comments. We hope that our response can clarify the misunderstandings and you can consider our work more favorably in the ratings.
>
> ### On the comparison to discrete diffusion techniques
> First, discrete diffusion methods can not allow  classifier-free guidance or gradient-based methods for conditional generation[1,2], which has been shown to be instrumental to the success of diffusion-based text-conditional image generators. In discrete methods, the corruptions are usually word masking/substitution in the discrete space. As a result, in the reverse process, it is infeasible to manipulate the controlling without the guidance of classifiers or perturb the hidden representation directly for controllable generation in the continuous space. Also, compared to discrete methods, continuous diffusion models provide more smoother transitions which provide smoother transitions between different states or levels of diffusion. This enables the examination of diffusion processes at any point in time, rather than being restricted to specific time intervals defined by discrete models. This allows better interpretation of the generation process as shown in our Table 6.
>
> Because of this, previous studies on continuous diffusion models [1,2] do not consider discrete diffusion models as baselines on conditional/controllable generation tasks. We follow [1] to mainly utilize well-known controllable generation methods together with the Diffusion-LM as our baseline to compare.
>
> Third, our experiments provide several insights to show the effectiveness of our proposed methods: (1) The main results in Table 1,3 as well as the inference time in Table 2 demonstrate the better performance and  efficiency compared to previous Diffusion-LM. Also, our methods can be better combined with LLMs. (2) Our ablation studies in Table 4 and 5 clearly demonstrate the effectiveness of the key design (soft-masked noise and objectives to predict the categorical distribution). (3) Our case studies further provide visualization and interpretation of the diffusion process where tokens are generated from an easy to hard manner.
>
> ### On the pre-trained model dependencies
> First, we would like to highlight that our methods boost the performance consistently with/without pre-trained models as shown in Table 1 in our paper in terms of automatic evaluations. We also emphasize the average rankings different methods receive from human evaluation (lower is better) in the table below (also in table 3). It is noteworthy that in both settings, our Masked-Diffusion-LM receives better ranks compared to Diffusion-LM. This suggests that our methods would improve the generation even without any pre-trained models. So our method does not depend on the use of pre-trained models.
> | Models              | BERT | Semantic Content | POS      | Syntax Tree | Syntax Spans | Length   |
> |---------------------|------|------------------|----------|-------------|--------------|----------|
> | Diffusion-LM        | No   | 2.89             | 2.76     | 3.16        | 2.88         | 2.46     |
> | Masked-Diffusion-LM | No   | **2.56**         | **2.48** | **2.88**    | **2.35**     | **2.18** |
> | Diffusion-LM        | Yes  | 3.87             | 3.46     | 3.72        | 3.68         | 3.34     |
> | Masked-Diffusion-LM | Yes  | **1.32**         | **1.28** | **1.16**    | **1.55**     | **1.86** |
>
> What’s more, in sharp contrast to previous work [1], our methods can be more effectively combined with pre-trained transformer models to utilize the advances in LLMs. Because we directly predict the categorical distribution instead of utilizing k-nearest-neighbor rounding technique to map continuous space to discrete space. This brings in better stability to handle high-dimensional representations from LLMs.
>
> [1] Li, Xiang, et al. "Diffusion-lm improves controllable text generation." Advances in Neural Information Processing Systems 35 (2022): 4328-4343.
>
> [2]Dieleman, Sander, et al. "Continuous diffusion for categorical data." arXiv preprint arXiv:2211.15089 (2022).

---

### Official Review · Reviewer_h85U · 2023-08-11

**Soundness:** 4

**Excitement:**

4: Strong: This paper deepens the understanding of some phenomenon or lowers the barriers to an existing research direction.

**Paper Topic And Main Contributions:**

The paper introduces a linguistic-informed soft-masking strategy in diffusion models for text generation, where important words would be perturbed with masks in an earlier stage. The proposed diffusion strategy combines discrete and continuous space and corrupts both of them during the diffusion forward process to model the textual data. The authors also explore the integration of pre-trained LM like BERT to further improve performance.

**Questions For The Authors:**

Q1: In the diffusion forward process, the paper indicates that the hidden representation of tokens is added by a small amount of Gaussian noise at each time step t. Are latent variables $\hat{X}_{t-1}$ corrupted by both Gaussian noise and MASK? An ablation study is needed to analyze their effects respectively.

Q2: In the loss function $\mathcal{L}_{t}$, the corrupted tokens are forced to learn two different targets, i.e. the original sentence $d$ and the masked sentence $\hat{d}$, which is an ambiguous and unstable target for the model.

Q3: When adapting BERT to the Masked-Diffusion LM, the parameters of BERT are frozen. Does it mean only the parameter of the final linear layer is updated or a new randomly initialized transformer (transition model) is built upon the BERT?

**Reasons To Accept:**

1. Well motivated. The hybrid diffusion model that makes corruptions in both discrete and continuous space is a novel method and worth further exploring.
2. Integrating pre-trained LM with diffusion models is a promising exploration.
3. The experiments comprehensively demonstrate the improvement of performance compared to baselines.

**Reasons To Reject:**

1. The paper lacks a description of the reverse process of the proposed soft-masking diffusion LM. The proposed method corrupts the textual data with both continuous Gaussian noise and discrete MASK tokens, which is different from typical diffusion models like DDPM. It is obviously impossible to directly apply the DDPM's reverse process to this method.

2. Non-bidirectional LMs like GPTs are not pre-trained with the Mask strategy, which is not compatible with the proposed soft-masking diffusion LM. The method could only be applied to LMs like BERT, which limits the possibility to scale up to LLM like GPT3.

**Reproducibility:**

2: Would be hard pressed to reproduce the results. The contribution depends on data that are simply not available outside the author's institution or consortium; not enough details are provided.

**Reviewer Confidence:**

5: Positive that my evaluation is correct. I read the paper very carefully and I am very familiar with related work.

---

> ### Author Rebuttal · Authors · 2023-08-28
>
> Thanks for your thoughtful comment.
>
> ### On the reverse process
> Intuitively, for our reverse process, different from DDPM, we directly predict the hidden representation of the target sequence at every step $t$, $X_{t-1} = f(X_{t})$ where $f(.)$ is a transition network such as a transformer, $X_{t}$ is the hidden representation of the sequence at step $t$. This can be viewed as predicting the representations of the tokens that are soft-masked at the step $t$ during the corruption process based on the recovered hidden representation by $f(.)$. And we further learn the feedforward layer $g(.)$ to predict the actual tokens based on the hidden representations $S_{t-1} = g(X_{t-1})$ to provide supervision to learn the transformer. We would elaborate more details on the reverse process in our revised version.
>
> ### On the application to Non-bidirectional LMs
> Our methods can be also applied to Non-bidirectional LMs such as GPT. As illustrated in Section 4.4 and Figure 1, we could just replace the encoder with GPT-based models to map the discrete tokens into continuous hidden representations. Then we applied the soft-masked corruption process and the reverse process to reconstruct the sequence. In the reverse process, we learn the transition network from a randomly initialized transformer based on the token predictions. Such a process does not involve any pre-trained transformers except the final linear layers to predict the tokens. As a result, our method does not rely on bidirectional LMs and can also be combined with autoaggressive LMs.
>
> ### On the questions about the forward process
> Our soft-masked noise can be viewed as a combination of both Gaussian noise and masking tokens. Specifically, in our forward process with soft-masked noise, compared to previous methods which add Gaussian noise to all the tokens representations at every time step, our method can be viewed as only adding a small amount of Gaussian noise to selected tokens at that time step (selected based on the word importance) which allows easy-first-generation nature in the reverse process. Also, instead of directly transforming the representations to 0s (the hard-masked noise adding to the discrete space) which create a large amount of noise at every time step, our soft-masking noise allows smaller corruption to the continuous space.
>
> ### On the questions about the loss function
> Our loss function at time step $t$ can be viewed as two parts: predicting the tokens that are not masked at $t$ and predicting the tokens that are masked at $t$. And we put higher weights on the later part. So the models would learn to predict the masked tokens first at every time step.
>
> ### On the questions about the learned parameter when adopting BERT
> When utilizing BERT, we freeze the pre-trained parameters in the BERT models and learn a new randomly initialized transformer which is built upon the BERT to serve as the transition model for the reverse process （recovering the corrupted hidden representations).

---

### Official Review · Reviewer_Q6XL · 2023-08-11

**Soundness:** 3

**Excitement:**

3: Ambivalent: It has merits (e.g., it reports state-of-the-art results, the idea is nice), but there are key weaknesses (e.g., it describes incremental work), and it can significantly benefit from another round of revision. However, I won't object to accepting it if my co-reviewers champion it.

**Paper Topic And Main Contributions:**

This paper proposes a new diffusion model for language modeling called Masked-Diffusion LM, where the authors designs a linguistic-informed soft masking noise process during the forward diffusion that gradually masks words from more informative to less informative.
For more efficient and stable training for text diffusion, they use cross-entropy loss to directly predict tokens instead of L2 loss on embeddings. The model also allows incorporating large pretrained language models like BERT by freezing BERT and learning the diffusion process.
Experiments are mainly conducted on controllable text generation showing the effectiveness of the proposed approach over Diffusion-LM

**Main contributions:**
Developing a diffusion LM that is tailored for language with soft masking and direct discrete prediction, and showing strong performance on controllable text generation tasks.

**Reasons To Accept:**

1. The linguistic-informed masking is intuitive and shows clear benefits in ablations. Gradual masking encourages generating simple words first.
2. Results show state-of-the-art performance on multiple controlled generation tasks compared to previous diffusion LMs and other methods.
3. The approach is compatible with pretrained LMs.


**Reasons To Reject:**

1. This paper only evaluate the proposed approach on controllable generation tasks. The paper would be strengthened by evaluating the proposed model in a multilingual setting, which has been commonly used in prior work on diffusion models for text generation. For example, several previous[1,2,3,5] studies have experimented with machine translation tasks like English-to-German translation on the WMT14 dataset. Applying the current model to translation or other multilingual datasets could further demonstrate its capabilities and allow for comparison to these prior methods. The lack of multilingual evaluation is a concern since the current benchmarks, while varied, are limited to monolingual English tasks.
2. The paper's comparisons to other relevant work could be expanded. In the experiments, only a couple recent diffusion models were compared against. Given significant recent progress in text diffusion modeling, more in-depth discussion of how the proposed model relates to other continuous diffusion methods (e.g. CDCD[1], Difformer[2], DINOISER[3]) and discrete diffusion methods (e.g. DiffusionBERT[4], RDM[5], DiffusER[6]) would strengthen the paper. The contributions could be better situated by clarifying the advantages of the current model compared to these other related studies.

---
[1] Continuous diffusion for categorical data

[2] Difformer: Empowering Diffusion Model on Embedding Space for Text Generation

[3] DINOISER: Diffused Conditional Sequence Learning by Manipulating Noises

[4] DiffusionBERT: Improving Generative Masked Language Models with Diffusion Models

[5] A Reparameterized Discrete Diffusion Model for Text Generation

[6] DiffusER: Discrete Diffusion via Edit-based Reconstruction

**Reproducibility:**

3: Could reproduce the results with some difficulty. The settings of parameters are underspecified or subjectively determined; the training/evaluation data are not widely available.

**Reviewer Confidence:**

4: Quite sure. I tried to check the important points carefully. It's unlikely, though conceivable, that I missed something that should affect my ratings.

---

> ### Author Rebuttal · Authors · 2023-08-28
>
> Thanks for your positive and constructive reviews.
>
> ### On the selection of the evaluation sets and testing multilingual datasets
> We mainly follow the previous work [1] to evaluate and compare different methods on 5 controllable generation tasks. In terms of the multilingual settings, we think our methods would still show improvements because: (1) our introduced strategical masking noise are based on the word relevance and entropy which are agnostic to language variations, and (2) our methods would also be easily to different pre-trained multilingual models to handle multilingual cases such as machine translations.
>
> ### On the comparison to other relevant work
> We have discussed the major comparison to previous discrete and continuous diffusion models in the language model domain [1,2,5] in the introduction and related work section: our work utilize soft-masking informed by linguistic features to corrupt both the discrete and continuous space, and then iteratively denoise them back by predicting the categorical distribution. Compared to discrete diffusion models like [5,6,7], we model the corruption process in the continuous space which enables flexible corruption and controllable reconstruction by manipulating the hidden representations which allow classifier-free guidance or gradients-based updates for conditional/controllable generation. And our soft-masking noise also shares the advantages of the discrete diffusion process to guide the corruption by linguistic structures (e.g., soft-masking tokens based on the importance. ). Compared to continuous diffusion models like [1,2,3,4], our methods have two major improvements: (1) we introduce a more structured noise guided by linguistic features that noise the discrete and continuous space at the same time instead of random Gaussian noise [1,2,3,4]; (2) we directly predict the categorical distribution during the denoising process to bridge the continuous and discrete space instead of rounding techniques [1,3] which allows faster convergence and the ability to combine large pre-trained models. We would like to expand these comparisons in our revised version.
>
> [1] Li, Xiang, et al. "Diffusion-lm improves controllable text generation." NeruIPS (2022).
>
> [2]Dieleman, Sander, et al. "Continuous diffusion for categorical data." arXiv preprint arXiv:2211.15089 (2022).
>
> [3]Gao, Zhujin, et al. "Difformer: Empowering diffusion model on embedding space for text generation." arXiv preprint arXiv:2212.09412 (2022).
>
> [4] Ye, Jiasheng, et al. "Dinoiser: Diffused conditional sequence learning by manipulating noises." arXiv preprint arXiv:2302.10025 (2023).
>
> [5] He, Zhengfu, et al. "Diffusionbert: Improving generative masked language models with diffusion models." ACL (2023).
>
> [6] Zheng, Lin, et al. "A reparameterized discrete diffusion model for text generation." arXiv preprint arXiv:2302.05737 (2023).
>
> [7] Reid, Machel, Vincent J. Hellendoorn, and Graham Neubig. "Diffuser: Discrete diffusion via edit-based reconstruction." ICLR (2023).

---

### Meta-Review · Area_Chair_siFA · 2023-09-27

**Recommendation:** 4

**Metareview:**

Although the reviewers acknowledge the intuitive masking approach, improved efficiency, and novelty, they also suggest improvements such as conducting multilingual evaluation, expanding comparisons to other diffusion models, and providing more details on the reverse process and target ambiguity.

---

### Decision · Program_Chairs · 2023-10-07

**Decision:**

Accept-Main

**Comment:**

Although the reviewers acknowledge the intuitive masking approach, improved efficiency, and novelty, they also suggest improvements such as conducting multilingual evaluation, expanding comparisons to other diffusion models, and providing more details on the reverse process and target ambiguity.